# Research on the Properties of Mineral–Cement Emulsion Mixtures Using Recycled Road Pavement Materials

**DOI:** 10.3390/ma14030563

**Published:** 2021-01-25

**Authors:** Łukasz Skotnicki, Jarosław Kuźniewski, Antoni Szydło

**Affiliations:** Roads and Airports Department, Faculty of Civil Engineering, Wroclaw University of Science and Technology, Wyb. Stanislawa Wyspianskiego 27, 50-370 Wroclaw, Poland; jaroslaw.kuzniewski@pwr.edu.pl (J.K.); antoni.szydlo@pwr.edu.pl (A.S.)

**Keywords:** complex modulus, shrinkage analysis, reclaimed asphalt, mineral–cement emulsion mixtures, cement dusty by-products (UCPPs)

## Abstract

The reduction in natural resources and aspects of environmental protection necessitate alternative uses of waste materials in the area of construction. Recycling is also observed in road construction where mineral–cement emulsion (MCE) mixtures are applied. The MCE mix is a conglomerate that can be used to make the base layer in road pavement structures. MCE mixes contain reclaimed asphalt from old, degraded road surfaces, aggregate improving the gradation, asphalt emulsion, and cement as a binder. The use of these ingredients, especially cement, can cause shrinkage and cracks in road layers. The article presents selected issues related to the problem of cracking in MCE mixtures. The authors of the study focused on reducing the cracking phenomenon in MCE mixes by using an innovative cement binder with recycled materials. The innovative cement binder based on dusty by-products from cement plants also contributes to the optimization of the recycling process in road surfaces. The research was carried out in the field of stiffness, fatigue life, crack resistance, and shrinkage analysis of mineral–cement emulsion mixes. It was found that it was possible to reduce the stiffness and the cracking in MCE mixes. The use of innovative binders will positively affect the durability of road pavements.

## 1. Introduction

The issue of environmental protection is very important, especially due to the progressive degradation and exploitation of the surrounding nature. For the construction of road pavements, not only new, unprocessed materials can be used but also those from recycling. Recycling used in road construction provides many tangible benefits: it reduces the need for mineral resources, lowers the cost of aggregate transport, and significantly reduces or even completely eliminates waste landfill from damaged road surfaces.

Recycling of road surfaces enables the reuse of road materials, which, after appropriate grading and mixing with binders such as asphalt or cement with their appropriate percentage, create full-value material products.

### 1.1. Protecting the Environment by Using Recycled Materials

For economic and ecological reasons, at the end of the last century in Europe, attempts were made to explain the problem of using building rubble as a building component. It was shown in [1] that the rubble from recycled masonry intended for the production of cement concrete should be precisely sorted. Despite the lower compressive strength, higher water absorption, and thus lower frost resistance of cement concrete containing aggregate derived from masonry, laboratory tests and experience in construction practice have clearly shown the suitability of recycled aggregate for the production of structural concrete also.

Aggregates recovered from the demolition of single-family houses, including foundations and walls, did not pose a threat to the environment when used in layers of road surfaces not bound with binder. As a mineral component, they did not meet the requirements specified in Spanish standards (grain size, Los Angeles abrasion), however, in the mineral mix, the recycled aggregates showed better structural behavior and less degradation and higher module values than the mix of natural aggregates. On the other hand, the durability was not lower than that obtained for natural aggregates [2].

Chinese experiments [3,4] confirmed the possibility of using concrete slag and brick slag for the construction of highway embankment surfaces. However, the humidity of the embankment should be controlled, as damaged or improperly drained pavement elements and slope inclinations may rapidly increase the humidity in the areas of water infiltration, and may also lead to its uncontrolled settlement. Construction and demolition waste (CDW) can be used in road embankments according to [5,6,7]. The use of demolition waste materials (CDW) in unbound layers of the foundation is confirmed by studies [8,9,10,11,12,13].

In Beijing, the authors of [14] analyzed the impact of construction waste on the function of the subgrade. Based on the determination of the degree of compaction, observation of settlement, and plate loading test, it has been shown that recycled CDW aggregates with appropriate sorting and appropriate construction technologies are useful. Based on the life cycle assessment, it was concluded that the use of recycled CDW can provide better environmental quality and economic benefits compared to direct disposal.

In [15], the results emphasize the possible technical benefits of using CDW materials instead of natural raw materials in applications such as roadside or unpaved roads. The monitored parameters were density, bearing capacity, and frost resistance. Assessment of the physical state and mechanical properties of CDW as waste from landfills [16] showed the possibility of using them as a base material.

Recycled mixes containing aluminum waste obtained better mechanical values (compressive strength, California bearing ratio (CBR) parameter) than the recycled mixes without this waste content [17].

The addition of metallic waste in the form of steel fibers to cement mortars resulted in a decrease in their bulk density and an increase in porosity [18]. Moreover, these metal wastes can modify the electrical resistance and thermal conductivity of the mortars, regardless of the type and amount of metal wastes. Such materials can be adopted, e.g., in self-de-icing road pavements where controlled temperature of the road layers is necessary.

In turn, the use of recycled fibers reduces the cost of the finished product by up to 50% in relation to the use of virgin fiber [19], without deteriorating the mechanical properties.

### 1.2. Waste Disposal—Asphalt, Rubber, and Concrete Reclaimed Waste

Chinese researchers [20] have shown in their research that reclaimed asphalt together with the recovered stabilized cement substrate can be used as a secondary raw material of aggregate for the preparation of cement-stabilized mixes in the cold recycling technology. Although the increase in the content of recycled materials resulted in a decrease in the value of mechanical properties, these mixes were characterized by sufficient durability and good performance.

The use of used tires for mixes with modified asphalt in pavement construction, according to [21,22], results in a large energy saving and reduction of carbon dioxide emissions. Such a solution provides many economic and environmental benefits, leading to the sustainable development of pavement infrastructure. Car rubber in the form of crumbs as a modifier improves the rheological and mechanical properties of rubber–asphalt mixtures [21]. Small pieces of rubber mixed with the soil obtain a beneficial effect in geotechnical engineering applications, such as providing better elastic deformations, improved shear strength, increased permeability, and better dynamic characteristics [23].

Increasing the reclaimed asphalt pavement (RAP) content in the mix reduces the strength of the mixtures but has no significant effect on drying shrinkage, erosion susceptibility, and capillary flow characteristics [24].

In Flanders (Belgium), up to 20% of coarse natural aggregates, as a component of concrete mixes, can be replaced with high-quality materials from recycling concrete aggregates [25].

In Malaysia [26], the results of the tests of intermediate tensile strength and modulus of elasticity showed that the addition of recycled concrete aggregate in the amount of 40% of the mineral mix to asphalt mixes is optimal and recommended.

In Hong Kong, a mix of recycled fine aggregate (RFA) and recycled concrete aggregate (RCA) [27] was used for construction applications using the coupled Taguchi–RSM optimization approach. Durability results confirmed the possibility of using this type of waste in building structures.

In [28], the authors showed that, for concretes with compressive strength up to 45 MPa, the type of aggregate used did not matter. It was based on several hundred designs of mixes with recycled concrete and natural aggregate.

Roughness and load-bearing capacity tests carried out on two sections of unpaved rural road showed no significant differences [29]. The surface layer of one section was made of a layer of recycled concrete aggregate and the surface layer of the other section was made of limestone aggregate. This type of waste can be used as an alternative to natural aggregates on unpaved country roads.

A significant influence of asphalt binders on physical and mechanical parameters, such as intermediate tensile strength, creep modulus, stiffness, or free space content, of mixes recycled using the cold method was demonstrated in the works [30,31]. The type of binder had a significant impact on the compaction properties of the tested mixes, which resulted in obtaining different values of mechanical parameters. The obtained results confirmed the possibility of applying cold recycled mixes with foamed asphalt to the layers of pavement structures [32].

### 1.3. Waste Utilization—Dusts

In Denmark [33], the environmental impact of stored ash and ash in road structures was estimated. The ecotoxicity in water showed a slight difference in their environmental impact. Dusts from electric furnaces do not deteriorate the strength parameter of asphalt concrete and at the same time improve its stiffness [34]. The use of dust as a substitute for Portland cement in the production of concretes dramatically reduces the compressive strength but results in lower sound permeability, which can be used for acoustic insulation [35]. Dust-containing mixes provide better thermal comfort, contributing to a reduction in operating energy in buildings.

Not only does the type of waste [36,37] used have an impact on the properties of road mixes, but the order and method of mixing and the conditions for mixing and compacting these components also have an impact on the obtained values of mechanical parameters, such as intermediate tensile strength or dynamic modules [38,39,40].

Road managers and policy makers should bear in mind the economic and environmental effects of mining rock raw materials, the demand for which is constantly growing. Only the configuration of an optimal model describing the production processes of obtaining rock raw materials or providing material data by producers [41,42] will enable the supply of aggregates in an economical and environmentally friendly manner and eliminate legal, technical, and market barriers [27].

### 1.4. Effect of Waste on Shrinkage and Stiffness

According to [43], it was noticed that the cracking sensitivity of mixes with recycled concrete decreased, despite a slightly lower tensile strength, compared to that of mixes containing natural gravel and sand. Moderate content of recycled aggregate (<30%) has little effect on tensile strength values, while complete replacement of natural aggregate with recycled materials reduces tensile strength by 20%.

Improvement of the properties of recycled mixes can be achieved not only by appropriate additives [44,45] but also thanks to “carbonization,” which improved mechanical properties and resistance to dry shrinkage for mixes containing recycled concrete aggregate (RCA) [46,47]. The carbonation process is relatively long but provides a more effective yield improvement.

It was observed that with the increase in the content of RFA and RCA, the contraction value increased [27,48], and with the increase of the water–cement ratio (WCR), the coefficient of the contraction value decreased. Similar observations are presented in [20]. The increase in the content of recycled aggregate resulted in an increase in optimal humidity, and its increase, in turn, increased the shrinkage value.

In [49], it was shown that cold recycled mix (CRM) achieved similar dry shrinkage values to cement stabilized macadam (CSM) mixes with only virgin aggregate. Higher cement content increases the dry shrinkage, which was also observed in [24]. As the temperature rises, the value of thermal shrinkage (TS) increases. CRM blends show worse TS resistance than CSM blends. Lower cement content is beneficial for the improvement of TS. Significant addition of coarse natural aggregate improves mechanical strength, frost resistance, resistance to dry shrinkage (DS), and CRM resistance to TS. However, the addition of fine aggregate makes CRMs more prone to shrinkage cracking, and freezing cycle failure [49]. The setting time is crucial in the initial phase; therefore, optimal humidity of the recycled layer must be maintained during this process to avoid high shrinkage on drying [24].

The properties of the mixes with the content of Portland cement were compared to mixes with lime-based low-carbon cementitious materials in [50]. It was found that not only the shrinkage value was lower in the mixes based on lime, but their strength increased after adding sodium sulfate to them. The RCA-containing blends achieved similar performance to the virgin aggregate blends through internal bonding caused by the release of moisture contained in the porous aggregate to allow further hydration of the cement [48]. Lower shrinkage was also observed in carbonized cement–slag mortars than in standard cement–slag mortars [46]. In [51], granulated blast furnace slag was replaced with fly ash and various types of fibers, and it was found that the addition of these materials did not adversely affect the setting time of the cement, slightly reduced strength, and reduced shrinkage compared to the slag mix. The mixes with fibers were characterized by increased strength and obtained the required resistance to freezing. Attempts were made to use kitchen waste oil and asphalt emulsion in mixes with recycled concrete aggregate stabilized with cement [52]. The results of the compressive and tensile strength tests showed that both additives caused an acceptable reduction in the strength of the mix in the early curing stage (7 days), but later, the treated mix showed a relatively fast increase in force to compensate for this deficiency, with a slightly higher value in the mix with the asphalt emulsion. These additives also reduced the stiffness of the mix and the amount of shrinkage. Moreover, the use of used oil and asphalt emulsion is ecologically friendly.

## 2. Materials and Methods

### 2.1. Purpose of the Research

The main objective of the research presented in this article is to assess the impact of an innovative binder on the physical, mechanical, and rheological properties of a recycled base layer made of mineral–cement emulsion (MCE) mixes. The result of this recognition is the development of a dedicated binder in the form of a hydraulic binder for the MCE substructure. The flowchart of the research approach is shown in Figure 1.

### 2.2. MCE Material as a Conglomerate for the Subbase Layers

The tests were performed for a total of 14 MCE blend variants, i.e., seven combinations for the fine-grained blend (MCE-D-1V, MCE-D-2V, MCE-D-3V, MCE-D-4C, MCE-D-5C, MCE-D- 6C, MCE-D-7C) and seven for coarse-grained mix (MCE-G-1V, MCE-G-2V, MCE-G-3V, MCE-G-4C, MCE-G-5C, MCE-G-6C, MCE-G-7C). The number of combinations equal to seven results from the plan of the simplex-centroid experiment [53,54]. Each point of the experiment describes the composition of the binder formed by combining the three basic components (cement, slaked lime, and cement dusty by-products). They differed from each other in the content of three components: cement, slaked lime, and cement dusty by-products (UCPPs). Cement dusty by-products are fillers made from wastes (blast furnace slags) obtained from heating plants. The test plan also used two reference mixes (MCE-D-Ref, MCE-G-Ref) using Portland cement as a binder.

The MCE mixtures are based on three main ingredients–aggregate, cement, and asphalt emulsion. The MCE mixtures are semi-rigid materials. When in the material there is only the cement, this material has high stiffness. After adding the asphalt emulsion and replacing part of the cement with cement dusty by-products (UCPPs), the reduction in stiffness and improvement in the fatigue durability of the road construction layer will be possible.

The compositions of the mixes were determined as assumed, i.e., for two types of mixes: fine-grained (marked with the symbol D) and coarse-grained (marked with the symbol G). The fine-grained mix consisted of natural grading aggregate with continuous graining 0/31.5 mm (melaphyre), natural grading aggregate with continuous graining 0/2 mm (sand), and recycled basalt aggregate with asphalt binder (recycled asphalt) 0/10 mm. On the other hand, the coarse-grained mixture contained natural grading aggregate with continuous graining 0/31.5 mm, natural grading aggregate with continuous graining 0/2 mm, and recycled basalt aggregate with asphalt binder (recycled asphalt) 0/31.5 mm.

This binder was designed to reduce stiffness and ensure service life for the MCE mix. The compositions of individual binders are presented in Table 1 and graphically in Figure 2 [54].

The adopted research plan in the field of physical, mechanical, and rheological properties was aimed at assessing the impact of the composition of the innovative binder on the change in the properties of the recycled base, which resulted in the development of nomograms (regression models) that were used to optimize the binder composition. The optimization made it possible to determine the most favorable configuration of the binder composition, taking into account the limitation of the stiffness of the foundation and ensuring the appropriate operational durability of the recycled foundation. The binder of the selected composition was then used to perform the experimental section, which enabled the assessment of its suitability in real conditions, i.e., as a binder in the foundation layer made of MCE in the construction of the road pavement.

### 2.3. The Scheme of the Recycled Substructure Composition

The design of the MCE mix should be correlated with the design of the pavement structure and the organization of works, depending on the method of its execution. The following materials were used to produce mineral–cement emulsion mixes: reclaimed asphalt, grading aggregate, hydraulic binder, asphalt emulsion, and water.

#### 2.3.1. Selection of the Optimal Grading Curve

The strength and deformation properties of MCE mixes are significantly influenced by the grain size composition of the mineral mix itself, in accordance with the requirements of the MCE mixes designing manual [55].

The conducted analyses show that the grading curve of the fine-grained and coarse-grained mineral mix meets the conditions of the MCE boundary curves and that the percentage of its individual components (natural aggregate and reclaimed aggregate) in both types of mixes is the same. In the design of mineral mixes, the percentage of asphalt waste was assumed on a level of 40%, natural aggregate 0/31.5 mm in the amount of 50%, and natural aggregate 0/2 mm in the amount of 10%—Table 2.

The grading of the mineral mixes was within the field of good grading—Figure 3 [55]. The grading of such a mineral mix was determined without taking into account the cement. The optimal percentage of reclaimed asphalt was adopted based on the analyses [54,56].

Both the fine and coarse mixes are grain continuous and comply with the boundary grading curves specified for MCE mixes intended for base layers.

#### 2.3.2. Designing the Optimal Amount of Hydraulic Binder (Cement) and Water

The mutual proportions between the individual components (recycled aggregate, material improving the gradation, hydraulic binder, and asphalt binder) determine the strength and deformation properties of MCE mixtures.

In the design of MCE mixes, based on own experience [57,58,59,60] and information recorded in world literature [52,61,62], the percentage of cement was assumed to be 3%. The designed MCE mixes use CEM II class 32.5 Portland cement.

#### 2.3.3. Designation of Optimum Moisture and Maximum Density of the Mix Skeleton

The optimal moisture content of the mix was determined by the Proctor method, based on the standard [63]. The optimal moisture and the maximum bulk density of the mix skeleton were determined.

From the tests [54,56], it was determined that for a fine-grained mix, the optimum humidity is 8.0%, with the maximum density of the mineral skeleton of the mixtures of 2.142 g/cm^3^. In turn, for a coarse-grained mixture, the optimal humidity is 7.8%, with the maximum density of the mineral skeleton of the mixes of 2.198 g/cm^3^.

##### Designing the Optimal Amount of Asphalt Emulsion

Having a specific cement content and knowing the optimum humidity of the mixes in question, it was possible to determine the percentage of asphalt emulsion. For the analyzed MCE mixes, the asphalt emulsion 60/40 (AE 60/40) was used in accordance with the requirements of the standard [64]. Additionally, according to the MCE mix designing guide [55], it is recommended that the emulsion meets the following conditions: asphalt type 50/70 or 70/100.

In the design of MCE mixes, using own experience [57,58,59,60] and knowledge taken from the world literature [65,66,67], the percentage of asphalt emulsion 60/40 at 5% was set. Such emulsion content allowed to obtain the total binder content in fine-grained mixtures at the level of 5.1%, and in coarse-grained mixes at the level of 4.9% (taking into account the binder contained in the emulsion and the destructor).

The amount of water added to the MCE resulted from the optimal moisture content of the mineral–cement mixes and the amount of water from the emulsion.

#### 2.3.4. The Composition of MCE Mixes

With the screening data prepared for all mineral materials in the MCE mix and after determining the appropriate content of cement, asphalt emulsion and water, the final design of the composition of the recycled fine-grained and coarse-grained MCE mixes was prepared. These mixes were called reference mixes. The percentages of individual components are presented in the Table 3 and Table 4.

The next step in the research was to determine the composition of MCE mixes, which instead of cement contained an innovative binder. Seven such binders were used, and they differed in their content of three components: cement, lime, and dust.

This binder was designed to reduce stiffness and ensure service life for the MCE mix. The compositions of individual binders are presented in Table 1.

### 2.4. Research

Mechanical properties for the established compositions of MCE mixes (16 mixes—14 with the innovative binder plus two with cement binder only) were determined. The use of an innovative cement binder in recycled materials resulted in the reduction of cracking in MCE mixes. In the research cycle, the following tests were taken into account:Tests of the complex module by the four-point bending test on prismatic materials (4PB-PR) method according to [68] at temperatures of −10, +5, +13, +25, and +40 °C;Module tests using the indirect tension test on cylindrical specimens (IT-CY) method according to [68] at temperatures of −10, +5, +13, +25, and +40 °C;Tests of shrinkage by the ring method according to [69] at a temperature of +25 °C.

#### 2.4.1. Reduction of Stiffness after Using Dusts Instead of Cement (Stiffness Tests)

Tests of the indirect tensile stiffness modulus, IT-CY, were carried out according to the standard [68]. Four cylindrical (Marshall) samples with a nominal diameter of 101.5 mm and a height of 63.5 mm were used for the analyses for each test and characteristic temperature. The tests were carried out at the following temperatures: −10, +5, +13, +25, and +40 °C. The samples were compacted using 2 × 75 blows in a Marshall automatic compactor (MULTISERW-MOREK, Brzeznica, Poland). Due to the presence of cement mixes and hydraulic binders in the composition, the curing period of the samples was 28 days. The view of the Marshall samples is shown in Figure 4.

During the tests, detailed registration of the loading vertical force and horizontal displacement was made, which was used to determine the stiffness modulus of the tested material—Equation (1).
(1)$$E_{IT-CY}=\frac{F\cdot (\nu +0.27)}{\left(z\cdot h\right)}$$
where *E_IT-CY_*—stiffness modulus (MPa), *F*—maximum vertical force (N), *ν*—Poisson’s ratio of the material (–), *z*—amplitude of the obtained deformation (mm), *h*—average thickness of the sample (mm).

The sample loading scheme is shown in Figure 5.

The tests of the complex module by the bending method of a four-point beam were performed according to the standard [68], and the fatigue life tests according to the standard [70]. The same test samples are used in both cases. Six samples in the form of bars with nominal dimensions of b = 60 mm, h = 50 mm, L_tot_ = 400 mm (according to the equipment’s limitation) were used for the analyses, for each test and characteristic temperature. The tests were carried out at the following temperatures: −10, +5, +13, +25, and +40 °C. The samples were compacted with a roller with the use of skid plates according to the standard [71] to obtain the volumetric density determined on the Marshall samples. Then, bars were cut from the compacted slab to the required nominal dimensions. Due to the presence of cement mixtures and hydraulic binders in the composition, the curing period of the samples was 28 days. The view of the sample in the form of a bar is shown in Figure 6.

During the tests with the method of bending a four-point beam, a detailed registration of the vertical loading force and the vertical displacement in the middle of the beam span was made. The obtained data were used to determine the complex modulus—Equation (2).
(2)$$E_{4PB-PR}=\sqrt{E_{1}{}^{2}+E_{2}{}^{2}}$$
where

*E*_4*PB*−*PR*_—dynamic complex modulus of MCE mixture (MPa),

*E*_1_—the real component—Equation (3),

*E*_2_—the imaginary component—Equation (4).
(3)$$E_{1}=\gamma \cdot \left(\frac{F}{z}\cdot \cos\left(\varphi \right)+\frac{\mu }{10^{6}}\cdot \omega^{2}\right)$$(4)$$E_{2}=\gamma \cdot \left(\frac{F}{z}\cdot \sin\left(\varphi \right)\right)$$
where *F*—load (kN), *z*—deflection in the middle of the sample’s span (mm), *φ*—phase angle (°), *ω*—loading frequency (rad/s), *μ*—mass factor (g), *γ*—form factor (mm^−1^).

In the case of testing the complex module, a constant load was applied in the form of a given micro-strain at the level of ε = 50 × 10^−6^ m/m with a frequency of 10 Hz. The sample loading scheme is shown in Figure 7.

#### 2.4.2. Cracking Reduction Caused by Shrinkage and Stiffness (Shrinkage Test)

Shrinkage analyzes of the tested materials were performed using the ring method according to [69] as for concrete conglomerates. For tests, a mixture of calibrated sand with cement binder is prepared, in which shrinkage is generated as a result of setting. The appropriate proportions of the ingredients (sand, binder, water) were determined on the basis of the consistency tests presented in [72].

Table 5 shows the w/s (water to binder) coefficients necessary to obtain slurries and mortars of standard consistency for individual binders. The consistency of the mortars was determined by the drop cone [73] and spreading method [74].

After the mortars were made, they were placed in the [69] ring. The method of compaction was vibrating for a period of 15 s. The test temperature was 25 °C. The curing periods of the samples were from 14 to 122 days. An example of how the ring is connected (view of the test apparatus) with an example of the test run is shown in Figure 8.

## 3. Results and Discussion

### 3.1. Different Methods for Stiffness Comparisons (4PB-PR and IT-CY)

On the basis of the obtained detailed results, the average values of the indirect tensile stiffness modulus, IT-CY, along with the standard deviations of the results were determined for all tested MCE mixes, divided into coarse-grained and fine-grained mixes—Figure 9 and Figure 10.

Based on the analyses of the test results, it was found that the test temperature had a significant impact on the IT-CY stiffness modulus of MCE mixes. The stiffness of the mixes decreases with increasing test temperature. Depending on the grain size and binder 1V, 2V, 3V, 4C, 5C, 6C, 7C, and cement (Ref) used, MCE mixes may have a different material specification. The use of innovative binders made it possible to reduce the stiffness of individual MCE mixes compared to that of reference mixes containing classic cement in the entire temperature range—Figure 11.

Depending on the binder used, both for fine-grained and coarse-grained mixtures, the decrease in the value of the modules was from 40% to 80% of the value of the module of the reference mixture with cement only. The values of modulus of fine-grained and coarse-grained mixes for mixes with the content of a given binder were similar.

The greatest reduction in stiffness was achieved by using 2V, 4C, and 7C binders, both in coarse-grained and fine-grained mixes.

Based on the detailed results obtained, the mean values of the complex modulus and fatigue life along with the standard deviations of the results were determined in the four-point beam test for all tested MCE mixes, divided into coarse-grained and fine-grained mixes—Figure 12 and Figure 13.

The obtained test results indicate that with the increase in test temperature, the values of the complex modulus 4PB-PR decrease. Both for coarse-grained and fine-grained mixes, the lowest values of modulus were observed for 2V and 4C mixes. In turn, the highest values of the complex modulus were obtained by the mixes of 3V and 5C. The same mixes achieved the maximum values of the IT-CY stiffness modulus. The use of innovative binders made it possible to reduce the stiffness of individual MCE mixes compared to that of reference mixes containing classic cement in the entire temperature range—Figure 14.

### 3.2. Correlation of the Results of 4PB-PR and IT-CY Tests—Conversion Factors

As a result of the analyses carried out and the observation of similar dependencies between the stiffness of the tested material obtained by the 4PB-PR and IT-CY methods, an attempt was made to obtain a correlation for these two types of modules, Figure 15. Such correlation will allow for the future use of these tests to estimate the durability of the pavement road layer in MCE technology.

Correlations in Figure 15 were set for different compositions of MCE mixtures (fine grained and course grained). The results are shown for all tested temperatures with application of different UCPPs (cement dusty by-products) content. In spite of different variables in the tests, obtained correlations can be set as linear functions with a high coefficient of determination, R. On the basis of the conducted analyses, a good correlation was obtained between the stiffness determined according to the IT-CY indirect tensile method and the stiffness of MCE mixes determined by the bending method of a four-point beam, 4PB-PR—Equation (5).
(5)$$E_{4PB-PR}=A\cdot E_{IT-CY}+B$$
where

*E*_4*PB*−*PR*_—dynamic complex modulus of MCE mixture (MPa),

*E_IT_*_−*CY*_—stiffness modulus of MCE mixture (MPa),

A—slope factor depending on the grain size of the mix, A = 0.2067–0.2681 (-),

B—directional form factor depending on the grain size of the mix, B = 584.9–702.65 (-).

These dependencies will allow the interchangeable application of the above-mentioned tests, which can be used in the fatigue life prediction of road surfaces, taking into account the same test conditions.

### 3.3. Shrinkage Analysis

The phenomenon of shrinkage is observed during curing of samples of mixes with cement binders. As a result, it leads to the appearance of cracks in the material—Figure 16.

In order to reduce the occurrence of shrinkage cracks, innovative waste binders were used instead of classic cement. The appropriate stiffness was maintained. In the shrinkage analyses, the cement binder was used as a reference against three innovative binders 1V, 2V, and 7C, which clearly differed in the properties demonstrated in the stiffness tests using the 4PB-PR and IT-CY methods. The tests were carried out at the temperature of 25 °C. The relationship between the curing time of the samples and the micro-contraction deformation is shown in Figure 17.

A significant decrease in destructive strains was observed after the use of innovative binders. At the same time, the curing time to destructive cracks was significantly longer.

As a result of the tests, it was possible to establish the relationship between the stiffness of the tested material and the shrinkage generated during its curing. The relationship between 4PB-PR stiffness and shrinkage is shown in Figure 18, while the relationship between IT-CY and shrinkage is shown in Figure 19. The analyses relate to the curing temperature of 25 °C. The minimum level of stiffness for MCE mixtures is 2000 MPa according to the criteria specified in [58]. This condition was met for the fine-grained mixture with the binder 7C.

Estimated levels of micro-strain for a minimum stiffness level of 4PB-PR = 2000 MPa are between −30 and −20 micro-strains. For such a range of shrinkage strains, it is possible to assume the IT-CY stiffness level—Figure 19.

As a result of the analyses, it was found that the minimum level of IT-CY stiffness modulus for MCE mixes should be in the range of 5000–6000 MPa. The obtained data made it possible to estimate the plane of stiffness of the tested recycled materials. As the stiffness increases, the probability of contraction cracks occurring in it increases. Micro-strain shrinkage (ms) dependency on IT-CY stiffness modulus and 4PB-PR complex modulus for fine-grained mix is presented in Figure 20. This dependency is shown in the Equation (6) with the coefficient of determination, R^2^ = 0.99.
(6)$$ms=19.8525-0.0134\cdot E_{4PB-PR}-0.0024\cdot E_{IT-CY}$$
where

*ms*—shrinkage of MCE mixture (µm),

*E_4PB_*_−*PR*_—dynamic complex modulus of MCE mixture (MPa),

*E_IT_*_−*CY*_—stiffness modulus of MCE mixture (MPa).

The use of materials from the central area of the plane in the road surface will allow for adequate durability of the layer with the maximum reduction of shrinkage cracks.

## 4. Practical Application of MCE Mixes

As part of the research on the use of recycled materials and the assessment of the actual suitability of waste in the form of cement by-products (UCPPs) in combination with reclaimed asphalt (RAP) for the structural layers of road pavements, an experimental section was made in the arrangement of layers of the flexible pavement structure. The thicknesses of the individual layers are shown in Table 6. The foundation uses MCE with a cement binder and MCE with an innovative type 5C binder.

The stiffness tests of the executed layers will be correlated with laboratory tests and will be used to estimate the fatigue life of the pavement layers. The results of these studies will be the subject of further publications.

## 5. Conclusions

The conducted analyses for MCE mixes (16 mixes—14 with the innovative binder plus two with cement binder only) showed the following:Coarse-grained and fine-grained mixes containing cement binder marked as MCE_G_Ref and MCE_D_Ref were adopted as reference MCE mixes. Then, instead of cement, innovative road binders were used, containing cement, hydrated lime, and cement dusty by-products (UCPPs). All mixes can be used to make the road surface foundation.The analyzes of the IT-CY stiffness modulus showed that the test temperature had a significant impact on the stiffness for MCE mixes. With increasing test temperature, the stiffness of the mixes decreased. The use of innovative binders allowed to reduce the stiffness of individual MCE mixes compared to that of reference mixes containing classic cement in the entire temperature range. The greatest reduction in stiffness was obtained after using 2V, 4C, and 7C binders both in coarse-grained and fine-grained mixes. The reduction in the value of the modules accounted for 40–80% of the value of the modulus of the reference mixture only with cement.Based on the analyses of the results of the 4PB-PR complex module, it was found that the value of this module was significantly influenced by the grain size distribution of the MCE mix. Fine-grained mixes were characterized by greater homogeneity and higher complex modulus as compared to coarse-grained mixes. Despite the microscopic cracking phenomenon, the MCE layer will be able to take loads and ensure adequate fatigue life of the entire road surface structure.With the increase in temperature, the percentage decrease in the values of the modules obtained by the 4PB-PR method was slightly lower than that determined by the IT-CY method. In the case of both described modules for fine-grained and coarse-grained mixes with the content of all the binders used for the tests, depending on their value on the temperature change, they maintained the same curvilinear course.As the stiffness of the material increases, the likelihood of shrinkage cracking increases. The research showed a good correlation between the stiffness test results and the shrinkage strains. The higher the stiffness of the material, the greater the shrinkage deformation. The use of innovative binders allowed to reduce shrinkage cracks and extend the time to maximum deformation. For a minimum accepTable 4PB-PR stiffness level of 2000 MPa, the shrinkage value is between −20 and −30 micro-strains. Further reduction of the material stiffness is not desirable due to its too low fatigue life [58,75]. For the determined level of deformation, it was possible to estimate the minimum stiffness level in the IT-CY method. The IT-CY stiffness modulus of mineral–cement emulsion mixes should be reduced as much as possible but not less than 5000–6000 MPa.The developed relationships between the stiffness of 4PB-PR and IT-CY will allow the maximum use of various types of research equipment in the process of designing and making MCE mixes in road surfaces.Observation of the experimental section and identification of the stiffness modules of the foundation layer with the MCE will allow the correlation of the results of the modules of the built-in layer and those determined in the laboratory for the same materials. The results of the work will be the subject of further publications.The demonstrated reduction of shrinkage cracks in MCE mixes, through the use of cement dusty by-products (UCPPs), allows for the conclusion that the layers of road pavements containing these binders will have a higher fatigue life compared to that of conventional solutions based on cement binder.The use of cement dusty by-products (UCPPs) and reclaimed asphalt (RAP) to make pavement layers can significantly reduce the amount of unprocessed waste that causes environmental degradation. At the same time, the use of recycled products allows reducing the consumption of natural resources in road construction.

## Figures and Tables

**Figure 1 materials-14-00563-f001:**
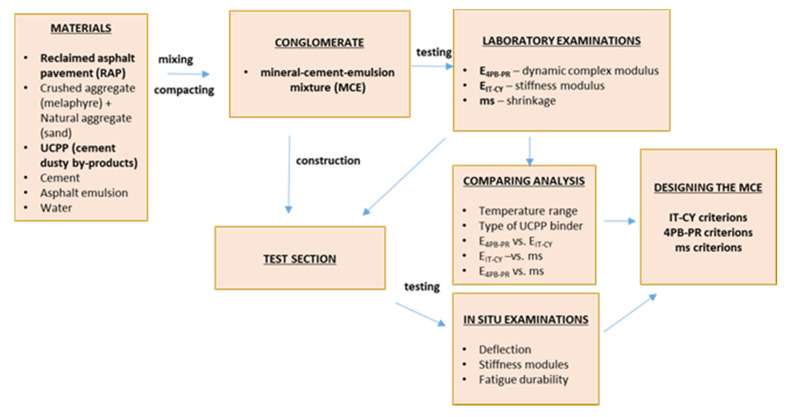
Research methodology flowchart.

**Figure 2 materials-14-00563-f002:**
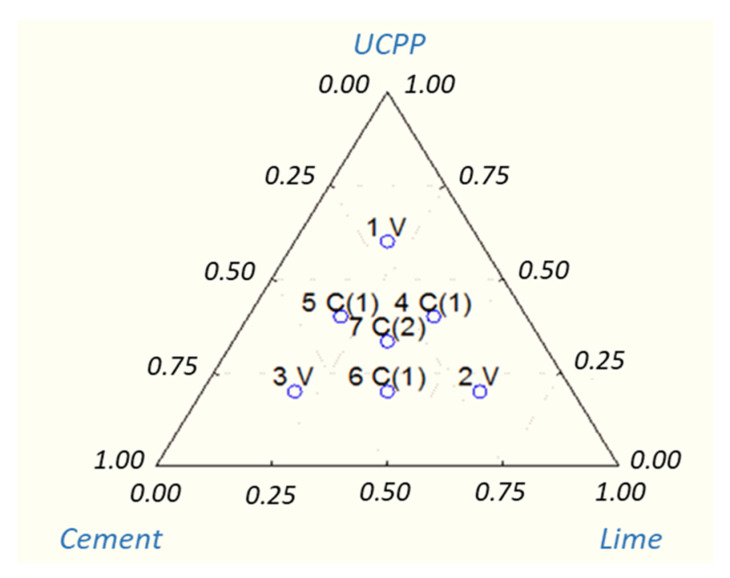
Plan of a simplex-centroid experiment [53,54].

**Figure 3 materials-14-00563-f003:**
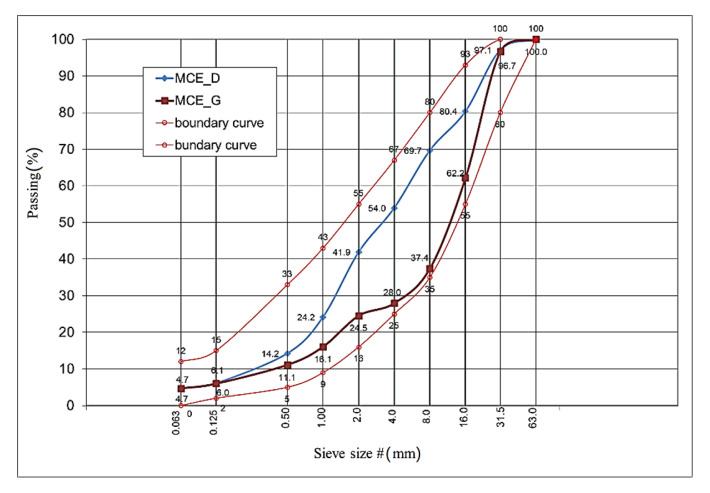
Mineral mixes’ grain size distribution and boundary curves.

**Figure 4 materials-14-00563-f004:**
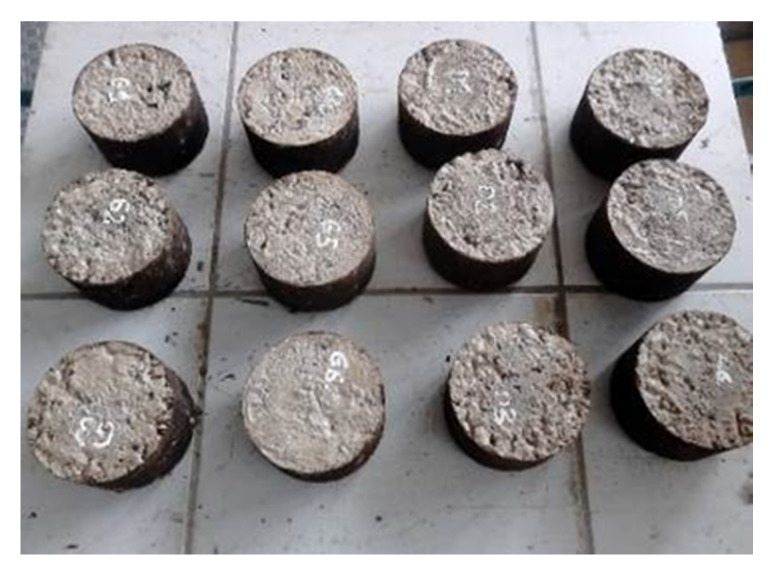
Marshall samples.

**Figure 5 materials-14-00563-f005:**
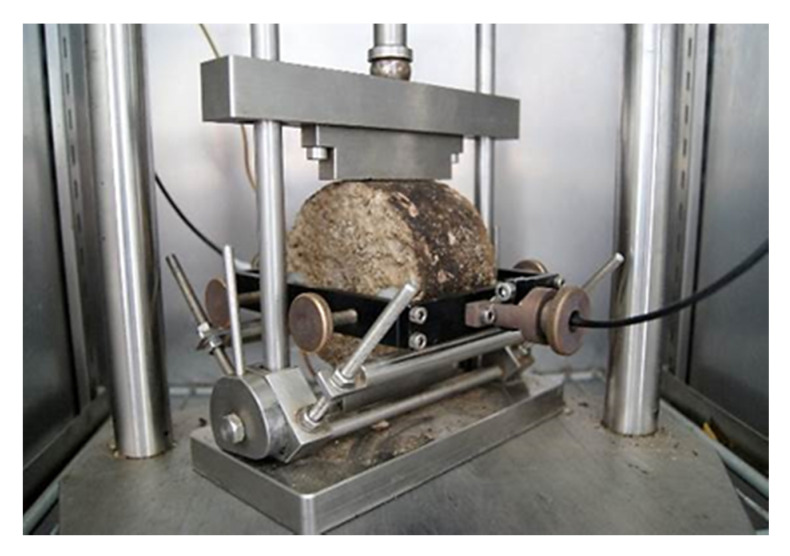
Scheme of the indirect tension test on cylindrical specimens (IT-CY) study.

**Figure 6 materials-14-00563-f006:**
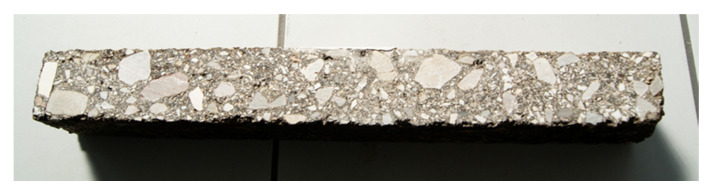
Sample for testing with the four-point bending test on prismatic materials (4PB-PR) method.

**Figure 7 materials-14-00563-f007:**
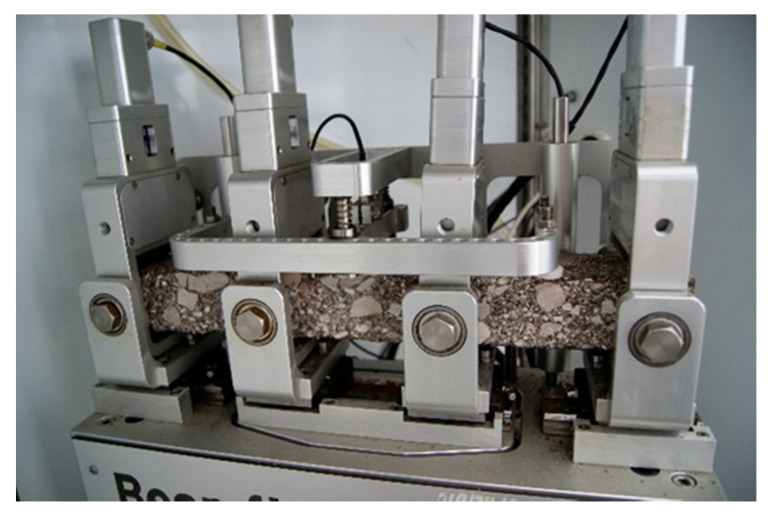
Scheme of the 4PB-PR test.

**Figure 8 materials-14-00563-f008:**
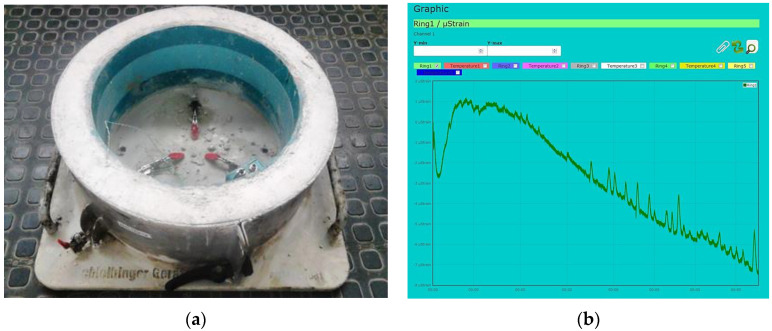
(**a**) View of the test apparatus and (**b**) the test run.

**Figure 9 materials-14-00563-f009:**
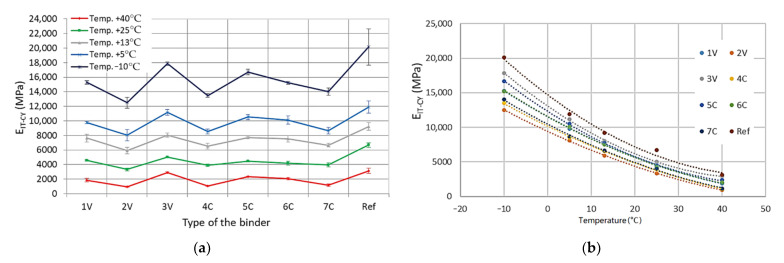
IT-CY stiffness modulus—coarse-grained MCE mix: (**a**) binder dependency, (**b**) temperature dependency.

**Figure 10 materials-14-00563-f010:**
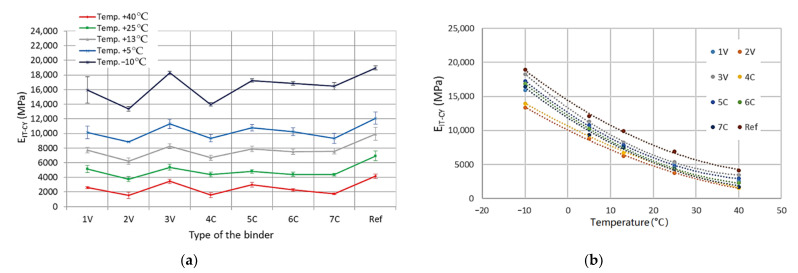
IT-CY stiffness modulus—fine-grained MCE mix: (**a**) binder dependency, (**b**) temperature dependency.

**Figure 11 materials-14-00563-f011:**
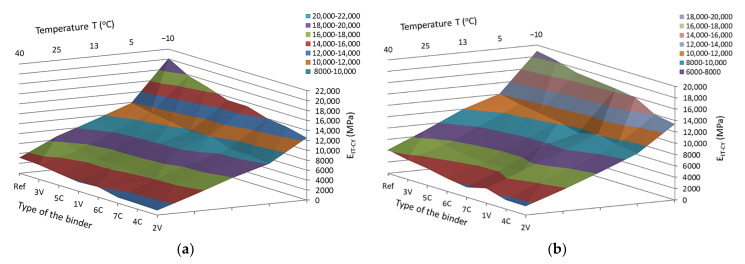
IT-CY module: (**a**) coarse-grained mixtures, (**b**) fine-grained mixtures.

**Figure 12 materials-14-00563-f012:**
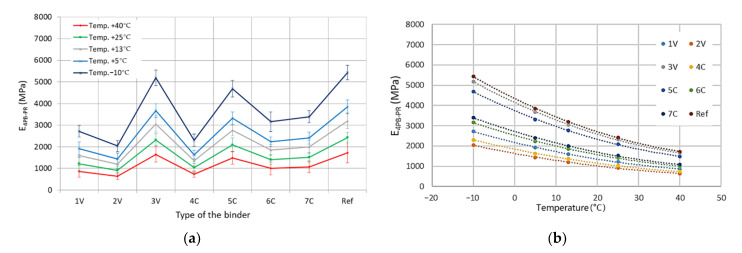
4PB-PR complex modulus—coarse-grained MCE mix: (**a**) binder dependency, (**b**) temperature dependency.

**Figure 13 materials-14-00563-f013:**
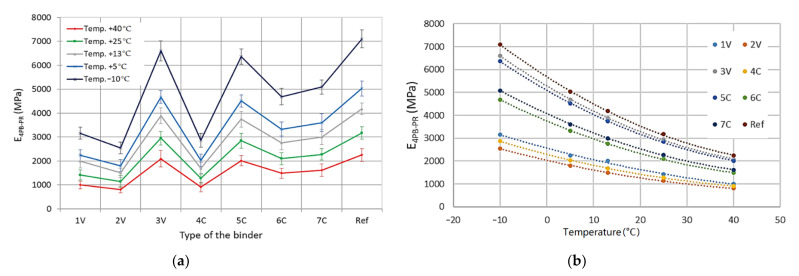
Complex modulus 4PB-PR—fine-grained MCE mix: (**a**) binder dependency, (**b**) temperature dependency.

**Figure 14 materials-14-00563-f014:**
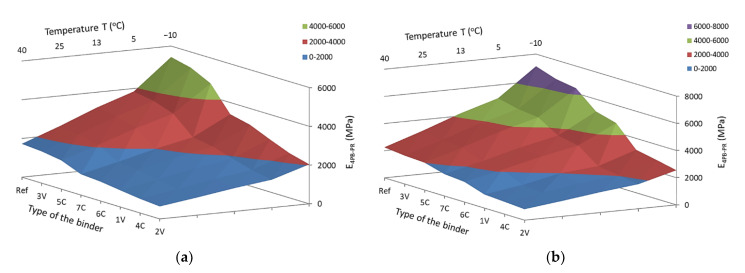
4PB-PR module: (**a**) coarse-grained mixes, (**b**) fine-grained mixes.

**Figure 15 materials-14-00563-f015:**
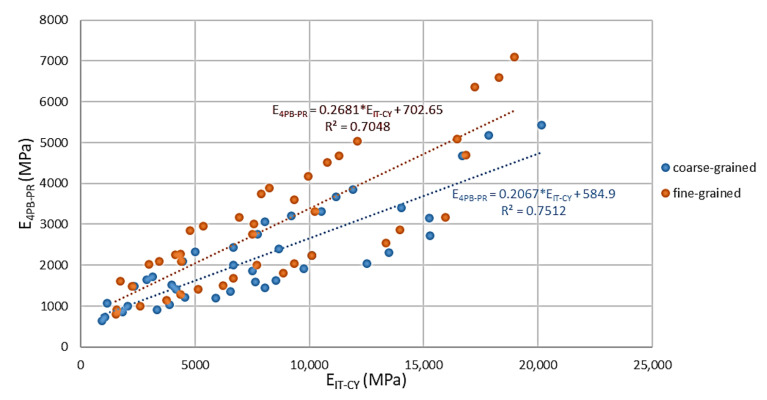
Correlation of modules obtained by 4PB-PR and IT-CY methods.

**Figure 16 materials-14-00563-f016:**
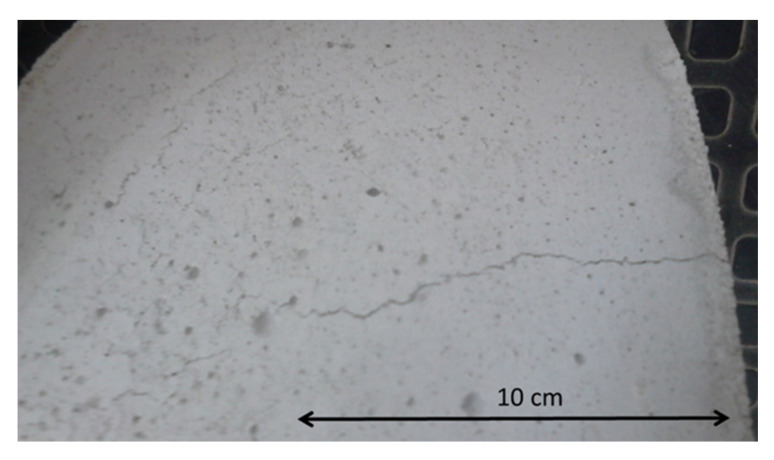
The occurrence of shrinkage cracks.

**Figure 17 materials-14-00563-f017:**
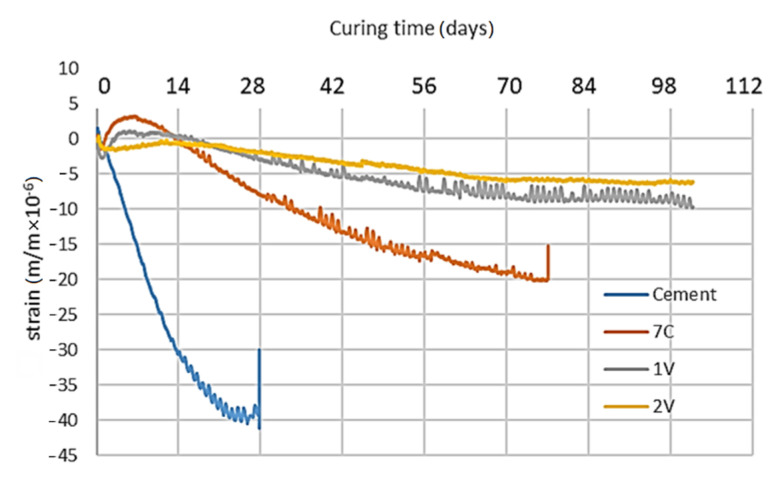
Increase in shrinkage micro-strain during material curing time.

**Figure 18 materials-14-00563-f018:**
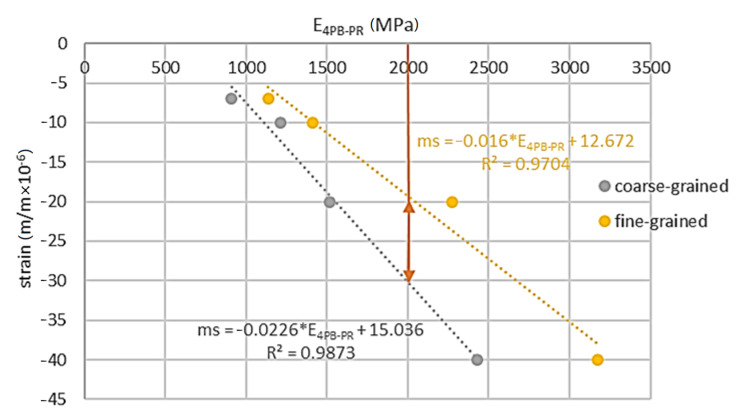
4PB-PR stiffness vs. shrinkage.

**Figure 19 materials-14-00563-f019:**
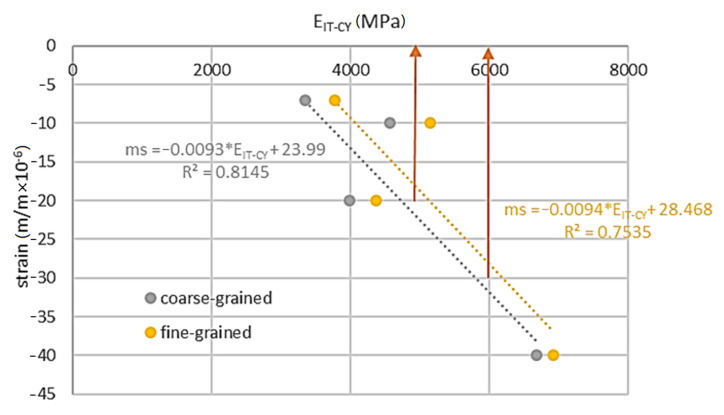
IT-CY stiffness vs. shrinkage.

**Figure 20 materials-14-00563-f020:**
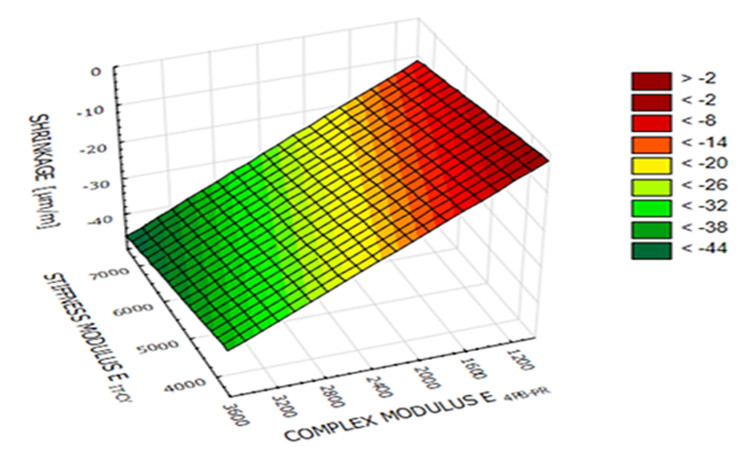
Stiffness plane of MCE mixes.

**Table 1 materials-14-00563-t001:** Designation and composition of the composed binders.

Symbol of Binder	Ingredient (–)		
	Cement	Lime	UCPP
1V	0.20	0.20	0.60
2V	0.20	0.60	0.20
3V	0.60	0.20	0.20
4C	0.20	0.40	0.40
5C	0.40	0.20	0.40
6C	0.40	0.40	0.20
7C	0.33	0.33	0.33
Ref	1	0	0

UCPP—cement dusty by-product.

**Table 2 materials-14-00563-t002:** Composition of mineral mixes.

Components of the Mineral Mix	Fine (MCE_D)	Coarse (MCE_G)
	(%)	(%)
0/10 reclaimed asphalt	40	-
0/31.5 reclaimed asphalt	-	40
0/31.5 crushed aggregate improving the gradation—melaphyre	50	50
0/2 natural aggregate improving the gradation—sand	10	10

MCE_D—fine-grained mineral–cement emulsion mixture; MCE_G—coarse-grained mineral–cement emulsion mixture.

**Table 3 materials-14-00563-t003:** Composition of the fine-grained mix.

Components	Mineral Mixture (MM)	Mineral–Cement Emulsion Mixture (MCE_Ref)
	(%)	(%)
0/10 reclaimed asphalt	40	34.4
0/31.5 crushed aggregate improving the gradation	50	43.0
0/2 natural aggregate improving the gradation	10	8.6
Cement	-	3.0
Asphalt emulsion 60/40	-	5.0
Water	-	6.0

**Table 4 materials-14-00563-t004:** Composition of the coarse-grained mix.

Components	Mineral Mixture (MM)	Mineral–Cement Emulsion Mixture (MCE_Ref)
	(%)	(%)
0/31.5 reclaimed asphalt	40	34.5
0/31.5 crushed aggregate improving the gradation	50	43.1
0/2 natural aggregate improving the gradation	10	8.6
Cement	-	3.0
Asphalt emulsion 60/40	-	5.0
Water	-	5.8

**Table 5 materials-14-00563-t005:** Ratios w/s (water to binder).

Binder	*w*/*s* for the Mortar
3 V	0.60
1 V	0.79
4 C	0.75
5 C	0.68
2 V	0.76
7 C	0.66
6 C	0.62

**Table 6 materials-14-00563-t006:** Layout of pavement construction layers used in the experimental section.

Layer Type	Material	Layer Thickness (m)
Wearing/base course layer	SMA (stone matrix asphalt) 0/16 mm	0.08
base layer	MCE and MCE + binder	0.20
natural subsoil		

## Data Availability

The data presented in this study are available on request from the corresponding author.

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
