# Peer review of "Research on the Properties of Mineral–Cement Emulsion Mixtures Using Recycled Road Pavement Materials"

_materials, 2021, doi:10.3390/ma14030563_

Round 1

Reviewer 1 Report

The paper is very intersting but some improvements should be done!

Correction, doubts and notes are provided along the text (PDF) to improve the reading.

In my opinion, the introduction is too large. The introduction subtitles are Ok but are somewhat disconnected to the respective text. Special focus should be done to the experimental working factors. This way, a better characterization of the products can be provided.

Correlation of modules obtained by 4PB-PR and IT-CY methods (figure 15) can be better discussed by identifying the samples and providing detailed analyses and respective compositions. Group sample analyses could explain high dispersion in some experiments (e.g between 2000 and 3500 4PB). Cement content and binder/water ratios seems to be very important.

In my opinion, overall photograph quality can be improved and detailed macros from contrasting properties are welcome!

In my opinion, less text and more flowcharts can improve paper´s reading.

In my opinion, chapter 4 is out of range. The paper is not a progressing report. A brief note can be presented at the end of the article but is not necessary!

Reviewer 2 Report

Congratulations to authors. Very interesting paper regarding the evaluation of the impact of an “innovative binder on the physical, mechanical and rheological properties of a recycled base layer made of mineral-cement-emulsion mixes”.

In general, the paper is well presented but some information shall be clarified or completed:

  1. It seems appropriate to include a list of acronyms (or “Abbreviations”) before “References”;
  2. Under the International System of Units recommendations (2019), a space must separate the number and the percentage symbol (“## %” and not “##%”);
  3. The same recommendation is made for Celsius degree symbol (“## °C” and neither “##°C” nor “##° C);
  4. Figure 3 and others: please remove all the hidden data (that appears when we place the cursor over the image);
  5. Line 155: what kind of lime was used in the cited paper [50]?
  6. Figure 2, Tables 3 to 5 and others: please change the decimal separator (dot and not comma);
  7. Why did the Authors change the “cationic emulsion C60B10 R” designation (line 285) by “Asphalt emulsion 60/40” as presented in Tables 3 and 4?
  8. Lines 286/287: the word “additionally” is used twice in the same paragraph;
  9. Line 325: is written “… 75 strokes per side …” but in some European Standards is commonly used the expression “… 2⨯75 blows …”;
  10. Line 343: the beams had “nominal dimensions of b = 60 mm, h = 50 mm, Ltot = 400 mm”. Taking into account that the maximum dimension of the aggregate in the mixture was 31,5 mm (D), the height h (50 mm) doesn’t respect the recommended value prescribed in the standard (EN 12697-26): the height h “should be at least three times the maximum grain size D in the tested material”. Please refer to equipment limitation;
  11. Line 361: complete with the units “… μ− mass factor [g], γ – form factor [mm-1].”;
  12. Line 363: please use the symbol “×” instead “*” in the expression “ε=50*10-6 m/m”;
  13. Figures 9 and 10: I would use the same scale in YY axes [for example, 24,000 MPa in both (a) and 25,000 MPa in both (b)];
  14. The same recommendation for Figures 12 and 13: I would use the same scale in YY axes (for example, 8,000 MPa in all cases);
  15. Lines 440/441: please note that the mentioned “interchangeable application” “which can be used in the fatigue life prediction of road surfaces” will only be relevant when using similar materials and the same test conditions;
  16. Figures 17 to 19: please verify the units of the YY axes (micro strain);
  17. Line 512: please complement with the lime type, “… containing cement, hydrated lime and cement dusty …”;
  18. The Authors may also include some information regarding one important aspect for the application industry: average costs of these solutions with innovative binders.

Reviewer 3 Report

  • I would suggest English edition of the manuscript. It is not extremely bad, but you use weird words order. It is hard to follow. 
  • You have not stated your objectives. It is hard to review a paper without clearly stated the purpose of the research.
  • I found some objectives in section 2.1. I think the objective part is very important and should be clearly stated in some separate section and not in some subsection. 
  • Line 258: Which specifications have you used to determine a good gradation?
  • Figure 2: For G, why have you left this flat piece of curve from 2 to 4 mm?
  • Why do you add asphalt emulsion and cement? What is the purpose? What would happen if you wouldn`t add emulsion? I think you should include some short discussion why you are using all of the components of your mixtures.
  • Figure 8: What do you mean by the "the course of the test?"
  • The conclusions section looks more like discussion. It is too long and you didn`t really concluded anything.

Round 2

Reviewer 1 Report

Considering that the authors carried out a profound restructuring
of the paper taking into account the proposed recommendations
and that this translates into greater legibility and quality of the text,
I suggest publication in the version presented.

Reviewer 3 Report

Thank you for addressing my comments.